# Plants Play Stronger Effects on Soil Fungal than Bacterial Communities and Co-Occurrence Network Structures in a Subtropical Tree Diversity Experiment

Huiyun Gan,[a,b] Xingchun Li,[a,b] Yonglong Wang,[c] Pengpeng Lü,[a,b] Niuniu Ji,[a,d] Hui Yao,[a,b] Shan Li,[e] ⬤Liangdong Guo[a,b]

aState Key Laboratory of Mycology, Institute of Microbiology, Chinese Academy of Sciences, Beijing, China
bCollege of Life Sciences, University of Chinese Academy of Sciences, Beijing, China
cFaculty of Biological Science and Technology, Baotou Teacher's College, Baotou, China
dDOE Center for Advanced Bioenergy & Bioproducts Innovation, University of Illinois at Urbana-Champaign, Urbana, Illinois, USA
eState Key Laboratory of Vegetation and Environmental Change, Institute of Botany, Chinese Academy of Sciences, Beijing, China

Huiyun Gan and Xingchun Li contributed equally to this article. Author order was determined by the corresponding author after negotiation.

**ABSTRACT** Increasing biodiversity loss profoundly affects community structure and ecosystem functioning. However, the differences in community assembly and potential drivers of the co-occurrence network structure of soil fungi and bacteria in association with tree species richness gradients are poorly documented. Here, we examined soil fungal and bacterial communities in a Chinese subtropical tree species richness experiment (from 1 to 16 species) using amplicon sequencing targeting the internal transcribed spacer 2 and V4 hypervariable region of the rRNA genes, respectively. Tree species richness had no significant effect on the diversity of either fungi or bacteria. In addition to soil and spatial distance, tree species richness and composition had a significant effect on fungal community composition but not on bacterial community composition. In fungal rather than bacterial co-occurrence networks, the average degree, degree centralization, and clustering coefficient significantly decreased, but the modularity significantly increased with increasing tree species richness. Fungal co-occurrence network structure was influenced by tree species richness and community composition as well as the soil carbon: nitrogen ratio, but the bacterial co-occurrence network structure was affected by soil pH and spatial distance. This study demonstrates that the community assembly and potential drivers of the co-occurrence network structure of soil fungi and bacteria differ in the subtropical forest.

**IMPORTANCE** Increasing biodiversity loss profoundly affects community structure and ecosystem functioning. Therefore, revealing the mechanisms associated with community assembly and co-occurrence network structure of microbes along plant species diversity gradients is very important for understanding biodiversity maintenance and community stability in response to plant diversity loss. Here, we compared the differences in community assembly and potential drivers of the co-occurrence network structure of soil fungi and bacteria in a subtropical tree diversity experiment. In addition to soil and spatial distance, plants are more strongly predictive of the community and co-occurrence network structure of fungi than those of bacteria. The study highlighted that plants play more important roles in shaping community assembly and interactions of fungi than of bacteria in the subtropical tree diversity experiment.

**KEYWORDS** bacteria, community assembly, co-occurrence network, fungi, plant species diversity, subtropical forest

Interactions between plants and soil microorganisms play a pivotal role in biodiversity maintenance, community stability, and ecosystem functioning (1, 2). Plants can affect soil microbial communities via host preference and changes in plant-derived inputs,

Address correspondence to Liangdong Guo, guold@im.ac.cn.

The authors declare no conflict of interest.

such as litter, rhizodeposits, and root exudates (3). In turn, soil microbes can influence plant diversity, productivity, and community composition through changes in soil available nutrients and the regulation of competitive interactions between plants (4). However, global change and human activity are causing increasing biodiversity loss, which profoundly affects community structure and ecosystem functioning (5, 6). Therefore, revealing the mechanisms associated with community assembly and co-occurrence network structure of microbes along plant species diversity gradients is very important for understanding biodiversity maintenance and community stability in response to plant diversity loss.

Previous studies demonstrated that soil fungal and bacterial community diversities and compositions could be affected by abiotic factors, such as soil, climate, and space in ecosystems (7–10). In addition, some studies indicated that soil bacterial species diversity and/or community composition were poorly or not predicted by plant species diversity in grassland (11–14), subtropical forest (15), and tropical forest (16) ecosystems, although few studies found significant relationships between them in grassland ecosystems (17, 18). In contrast, several findings have illustrated that plant species diversity was significantly related to soil fungal diversity and/or community composition in grassland (11, 19–24), oak savannas (25), temperate forest (26–28), subtropical forest (29), and tropical forest (30) ecosystems, even though several inconsistent results were found in grassland (12–14) and subtropical forest (15) ecosystems. Soil fungi may be more closely associated with plants than bacteria (mainly saprotrophic) because some fungi can form biotrophic interactions with trees in the form of root symbionts, pathogens, and endophytes (29–32). Additionally, fungi rely more heavily on plant-derived nutrients because fungal decomposers can decompose lignin and cellulose in litter and root exudates via extracellular enzymes (33, 34). In contrast, soil bacteria preferentially utilize labile organic products released from complex organic substrates (35, 36), while some bacteria, such as *Nocardia*, *Rhodococcus*, and *Streptomyces viridosporus*, were able to break down lignin (37–39). Therefore, plants may have a greater impact on the soil fungal community than on the bacterial community.

Disentangling the interactions among co-occurring organisms using ecological network analysis could provide new insights into the mechanisms underlying species' coexistence and community stability (40, 41). In microbial co-occurrence networks, most previous studies have focused mainly on fungi (42, 43) or bacteria (44–47) in ecosystems. However, only a few studies have compared the structure and potential drivers of soil fungal and bacterial co-occurrence networks simultaneously (48–50). For instance, de Vries et al. (50) revealed that the soil fungal co-occurrence network was less connected but more modular than the bacterial co-occurrence network during drought in a grassland. In another study, the co-occurrence networks of soil fungi and bacteria were found to exhibit higher edge numbers and degree centralization in the northern region than in the southern region in natural forests across eastern China, and geographic distance, climate, and soil properties were significantly related to the topological features of bacterial and fungal co-occurrence networks (48). Nevertheless, the co-occurrence network structure and potential drivers of soil bacteria and fungi in relation to plant species diversity in subtropical forest ecosystems are poorly documented.

Subtropical forests are widely distributed across South and East China, support a high diversity of plants (51) and soil bacteria and fungi (29, 52), and make major contributions to ecosystem services, such as carbon (C) cycling and terrestrial gross primary production (53). However, global environmental change and human activity are causing increasing biodiversity loss, which profoundly affects the ecosystem structure and functions (5, 6). To understand the relationship between plant diversity and ecosystem functions, a biodiversity-ecosystem functioning experiment was established in a Chinese subtropical forest (54), and studies found that increasing plant diversity strongly promoted plant stand-level productivity (55), functional diversity (56), above ground and belowground C storage (57), and herbivore phylogenetic diversity (58). In addition, plant diversity had a significant effect on the soil and root fungal communities (59, 60) and the specialization and

modularity of the tree-fungus bipartite network (61). However, the differences in community assembly and potential drivers of the co-occurrence network structure of soil fungi and bacteria in the subtropical tree diversity experiment are largely unknown.

To reveal the underlying mechanisms affecting the community assembly and co-occurrence network structure of soil bacteria and fungi along a plant species diversity gradient, we examined soil fungal and bacterial communities in a Chinese subtropical tree species richness experiment (1 to 16 species) using amplicon sequencing targeting the internal transcribed spacer 2 (ITS2) and V4 hypervariable region of the rRNA genes. Because soil fungi may be more closely associated with plants than bacteria, we hypothesize that (1) plants are more strongly predictive of the diversity and community composition of fungi than those of bacteria, and (2) plants play a more important role in shaping fungal than bacterial co-occurrence network structures in the subtropical tree diversity experiment.

## RESULTS

**General characterization of Illumina sequencing data.** After controlling for sequence quality, 3,249,819 ITS2 and 3,059,615 16S sequences were obtained from 3,897,827 and 3,852,057 raw sequences and clustered into 6,112 and 6,986 operational taxonomic units (OTUs) at a 97% similarity level, respectively. Among the 6,112 ITS2 OTUs, 5,840 were identified as fungal. The fungal sequence number was further rarefied to 15,366 (15,366 to 59,618 sequences in all the soil samples), resulting in a rarefied data set containing 5,669 fungal OTUs (1,090,986 sequences). Among the fungi, 5,235 OTUs (93.7% of total fungal sequences) were identified into 16 phyla and 434 OTUs into unidentified fungi (6.3%) (Table S1). The fungal community was dominated by Ascomycota (2,596 OTUs, 52.8% of total fungal sequences) and Basidiomycota (1788 OTUs, 27.4%) (Fig. S1A). Among the 6,986 16S OTUs, 6, 556 were identified as bacterial. The bacterial sequence number was rarefied to 10,723 (two samples with 2,631 and 7,019 sequences were discarded in this step), resulting in a rarefied data set containing 6,181 bacterial OTUs (739,887 sequences). Of the bacteria, 6,038 OTUs (99.8% of total bacterial sequences) were identified into 34 phyla and 143 OTUs into unidentified bacteria (0.2%) (Table S2). The bacterial community was dominated by Acidobacteriota (420 OTUs, 31.8% of total bacterial sequences) and Proteobacteria (1,123 OTUs, 24.6%) (Fig. S1B). For both fungi and bacteria, rarefaction curves of the observed OTU richness rose continuously with increasing sample numbers at the different levels of tree species richness, suggesting that further sampling would recover more OTUs (Fig. S2).

**Communities of fungi and bacteria.** The OTU richness of fungi and bacteria ranged from $608.5 \pm 128.6$ to $674.2 \pm 102.3$ and $629.4 \pm 215.8$ to $741.7 \pm 150.9$ (means $\pm$ SD), respectively. The Shannon diversity index of fungi and bacteria ranged from $4.352 \pm 0.785$ to $4.770 \pm 0.242$ and $5.480 \pm 0.214$ to $5.610 \pm 0.190$, respectively. The Simpson diversity index of fungi and bacteria ranged from $0.925 \pm 0.099$ to $0.968 \pm 0.095$ and $0.991 \pm 0.001$ to $0.992 \pm 0.001$, respectively. The linear model result indicated that the OTU richness, Shannon diversity index, and Simpson diversity index of fungi and bacteria were not significantly related to tree species richness and volume and soil variables (Fig. 1; Table S3).

Permutational multivariate analysis of variance (PerMANOVA) showed that tree species richness had a significant effect on the community composition (Bray-Curtis dissimilarity) of fungi ($R^2 = 0.020$, $P = 0.027$) but not bacteria ($R^2 = 0.018$, $P = 0.121$) (Fig. 2). Furthermore, pairwise PerMANOVA and nonmetric multidimensional scaling (NMDS) ordination indicated that the fungal community composition of monocultures was significantly different from that of 4-species, 8-species, and 16-species mixtures, and the fungal community composition of 2-species was significantly different from that of 4-species, and 16-species mixtures, but no significant difference in bacterial community composition among the five tree species richness levels was observed (Fig. 2; Table S4).

Variation partitioning showed that 15.7% of the variation in fungal community composition (Bray-Curtis dissimilarity) was explained by spatial distance (9.2%), soil (5.9%), tree

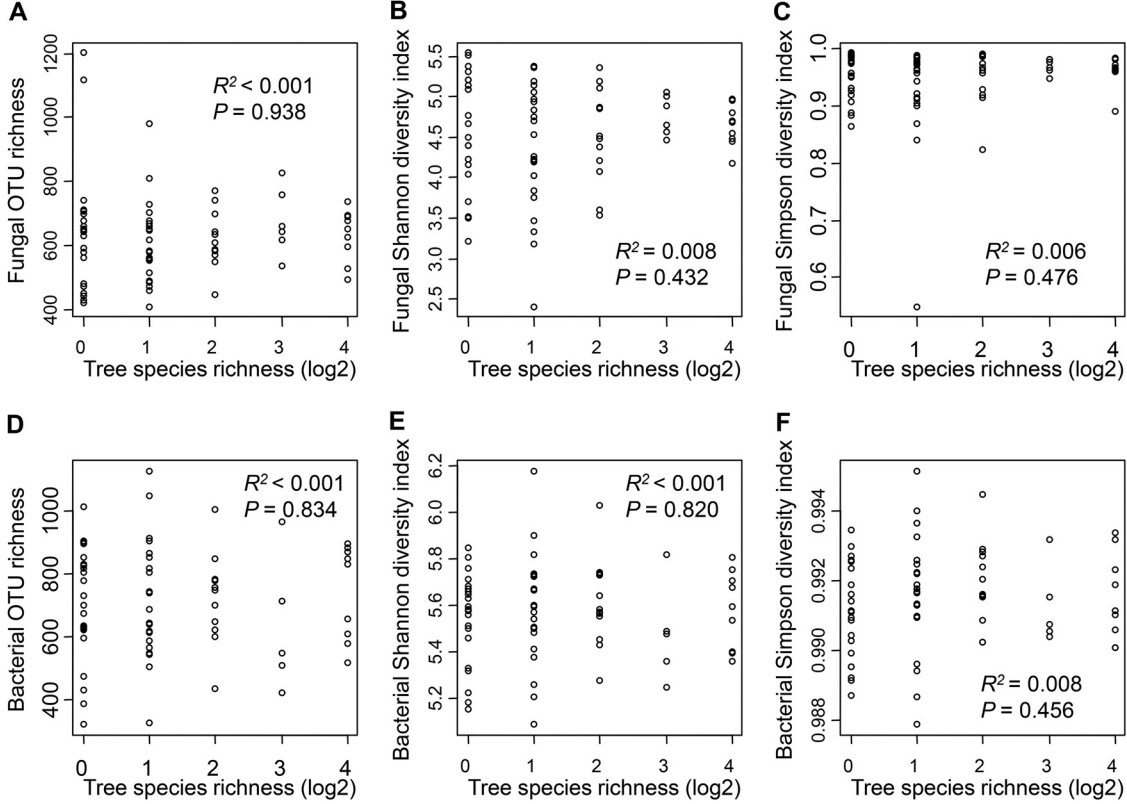

**FIG 1** Linear regression models showing the relationships between fungal and bacterial richness, Shannon diversity index and Simpson diversity index, and tree species richness classes. (A to C) Fungi. (D to F) Bacteria. OTU, operational taxonomic unit.

community (5.2%), and tree species richness (1.3%), with corresponding pure effects of 4.9%, 3.1%, 2.3%, and 1%, respectively (Fig. 3A). In contrast, 18.7% of the variation in bacterial community composition was explained by spatial distance (12.6%) and soil (10.8%), with corresponding pure effects of 7.8% and 6%, respectively (Fig. 3B). Furthermore, hierarchical partitioning analysis showed that 12.5% of the variation in fungal community composition (Bray-Curtis dissimilarity) was explained by soil (4.7%), spatial distance (4.0%), tree community (3.2%), and tree species richness (0.7%), with corresponding pure effects of 2.0%, 3.2%, 3.1%, and 0.8%, respectively (Fig. 4 and Table S5). In contrast, 12.9% of the

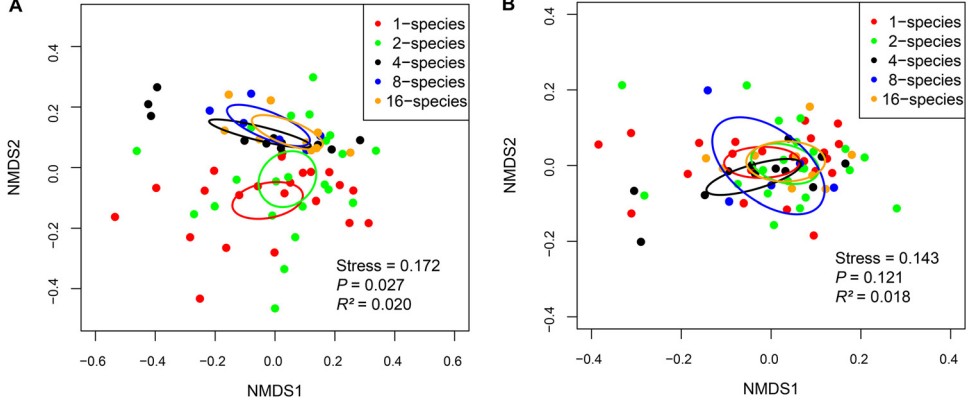

**FIG 2** Nonmetric multidimensional scaling (NMDS) ordination of the community composition (Bray-Curtis dissimilarity) of fungi and bacteria. (A) Fungi. (B) Bacteria. Ellipses in the plots denote 95% confidence intervals for the centroids of tree species richness. Permutational multivariate analysis of variance (PerMANOVA) showed that tree species richness had a significant effect on the community composition of fungi ($R^2 = 0.020$, $P = 0.027$) but not bacteria ($R^2 = 0.018$, $P = 0.121$).

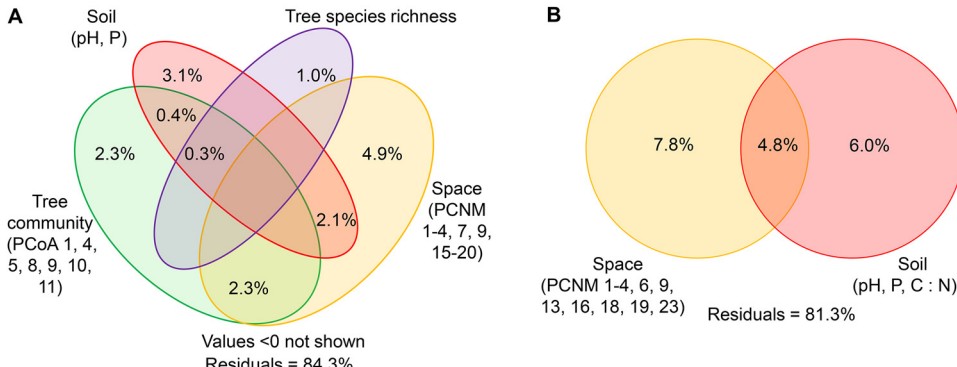

**FIG 3** Variation partitioning analysis showing the pure and shared effects of plant and abiotic factors on the community composition (Bray-Curtis dissimilarity) of fungi and bacteria. (A) Fungi. (B) Bacteria. Numbers indicate the proportion of explained variation. PCoA, principal coordinate analysis for the tree community; PCNM, spatial principal coordinates of neighbor matrices; P, soil total phosphorus; C, soil total carbon; N, soil total nitrogen.

variation in bacterial community composition was explained by soil (8.9%) and spatial distance (4.0%), with corresponding pure effects of 5.9% and 2.3%, respectively (Fig. 4 and Table S5). In addition, similar results were found in the PerMANOVA, NMDS, variation partitioning, and hierarchical partitioning analyses using Aitchison matrices of bacterial and fungal communities (Fig. S3 to S5, Table S6 and S7). These results indicate that plants exert a stronger effect on fungal than bacterial community compositions, while low variations of fungal and bacterial communities were explained.

**Co-occurrence networks of fungi and bacteria.** We produced a fungal co-occurrence network consisting of 72 nodes (OTUs) and 70 links; in contrast, the bacterial co-occurrence network consisted of 104 nodes and 264 links (Table 1). In the fungal and bacterial co-occurrence networks, the proportion of positive links was 1 and 0.864, the average degree 2.057 and 5.077, the clustering coefficient 0.566 and 0.370, the degree centralization 0.101 and 0.193, and the modularity 0.788 and 0.401, respectively (Table 1). The degree values for the fungal and bacterial co-occurrence networks fol-

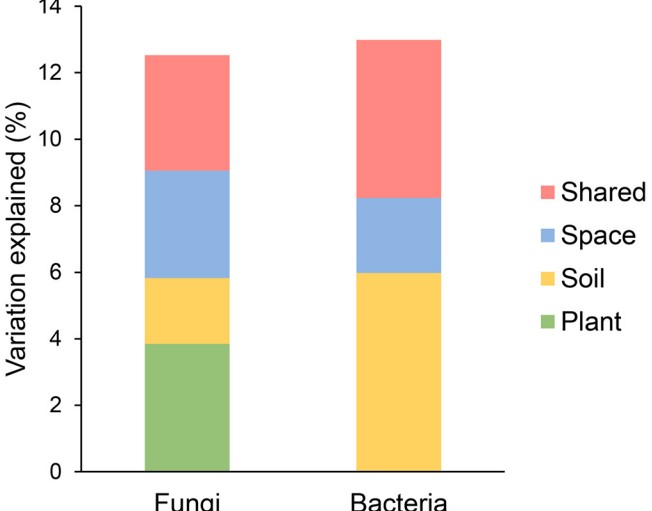

**FIG 4** Hierarchical partitioning analysis showing the pure and shared effects of plant and abiotic factors on the community composition (Bray-Curtis dissimilarity) of fungi and bacteria. Results from hierarchical partitioning analysis, aimed at identifying the percentage variance of the community composition of fungi and bacteria explained by the plant (tree species richness and community composition), space, and soil variables. Pure and shared variance from the plant, space, and soil variables in predicting the community composition of fungi and bacteria are merged in this figure for simplicity. An alternative version of this figure showing the pure and shared variance of each predictor can be found in Table S5.

**TABLE 1** The topological features of the fungal and bacterial empirical and random co-occurrence networks[a]

| Topological feature | Fungi | | Bacteria | |
| --- | --- | --- | --- | --- |
| | **Empirical network** | **Random network** | **Empirical network** | **Random network** |
| No. of links | 72 | 72 | 264 | 264 |
| No. of nodes | 70 | 70 | 104 | 104 |
| Positive links | 72 | | 228 | |
| Negative links | 0 | | 36 | |
| Proportion (Positive/total) | 1 | | 0.864 | |
| Average degree | 2.057 | 2.057 | 5.077 | 5.077 |
| Clustering coefficient | 0.566 | 0.029 ± 0.023 | 0.370 | 0.049 ± 0.010 |
| Degree centralization | 0.101 | 0.056 ± 0.013 | 0.193 | 0.060 ± 0.012 |
| Modularity | 0.788 | 0.645 ± 0.029 | 0.401 | 0.394 ± 0.012 |

[a]The Erdös–Réyni random networks were allocated the same number of nodes and edges as the corresponding co-occurrence networks. The topological features of random networks were calculated as the average value from 1,000 Erdös–Réyni random networks; Data are average value ± standard deviation.

lowed a power-law distribution ($R^2$ = 0.994 and $R^2$ = 0.971, respectively; Fig. S6), suggesting a scale-free network structure. The topological features of fungal and bacterial co-occurrence networks were greater than those of the Erdös-Réyni random networks (Table 1), indicating a nonrandom co-occurrence pattern and a small-world topology.

Variation partitioning indicated that 40.8%, 49.0%, 30.2% and 37.2% of the variances in average degree, clustering coefficient, degree centralization and modularity of the fungal co-occurrence network were explained by tree community (26.7%, 34.1%, 22.9% and 37.4%), tree species richness (27.0%, 31.4%, 13.7% and 14.1%) and soil (16.3%, 20.5%, 10.3% and 7.6%) (Fig. 5A, Fig. S7A to D). In contrast, 42.3%, 16.5%, 12.7% and 29.4% of the variances in average degree, clustering coefficient, degree centralization and modularity of the bacterial co-occurrence network were explained by soil (28.1%, 5.0%, 6.9% and 19.7%) and spatial distance (28.9%, 12.9%, 7.7% and 19.4%) (Fig. 5B, Fig. S7E to H).

The number of nodes and links in the fungal co-occurrence network, but not the bacterial co-occurrence network, generally decreased with increasing tree species richness (Fig. 6A and B; Table S8). Furthermore, correlation analysis showed that the average degree, degree centralization, and clustering coefficient in the fungal co-occurrence network significantly decreased but the modularity significantly increased with increasing tree species richness (Fig. 6C to F). However, no significant correlations

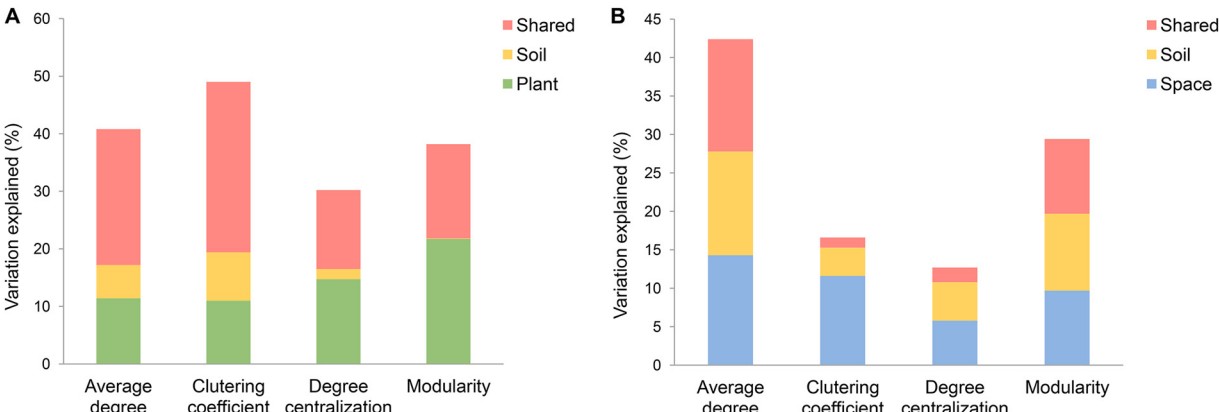

**FIG 5** Relative contribution of the variables to determining the topological features of the fungal and bacterial co-occurrence networks. (A) Fungi. (B) Bacteria. Results from variation partitioning modeling, aimed at identifying the percentage variance of the topological features of the fungal and bacterial co-occurrence networks explained by the plant (tree species richness and community composition), space, and soil variables. Unique and shared variance from the plant, space, and soil variables in predicting the topological features of fungal and bacterial co-occurrence networks are merged in this figure for simplicity. An alternative version of this figure showing the unique and shared variance of each group of predictors can be found in Fig. S8.

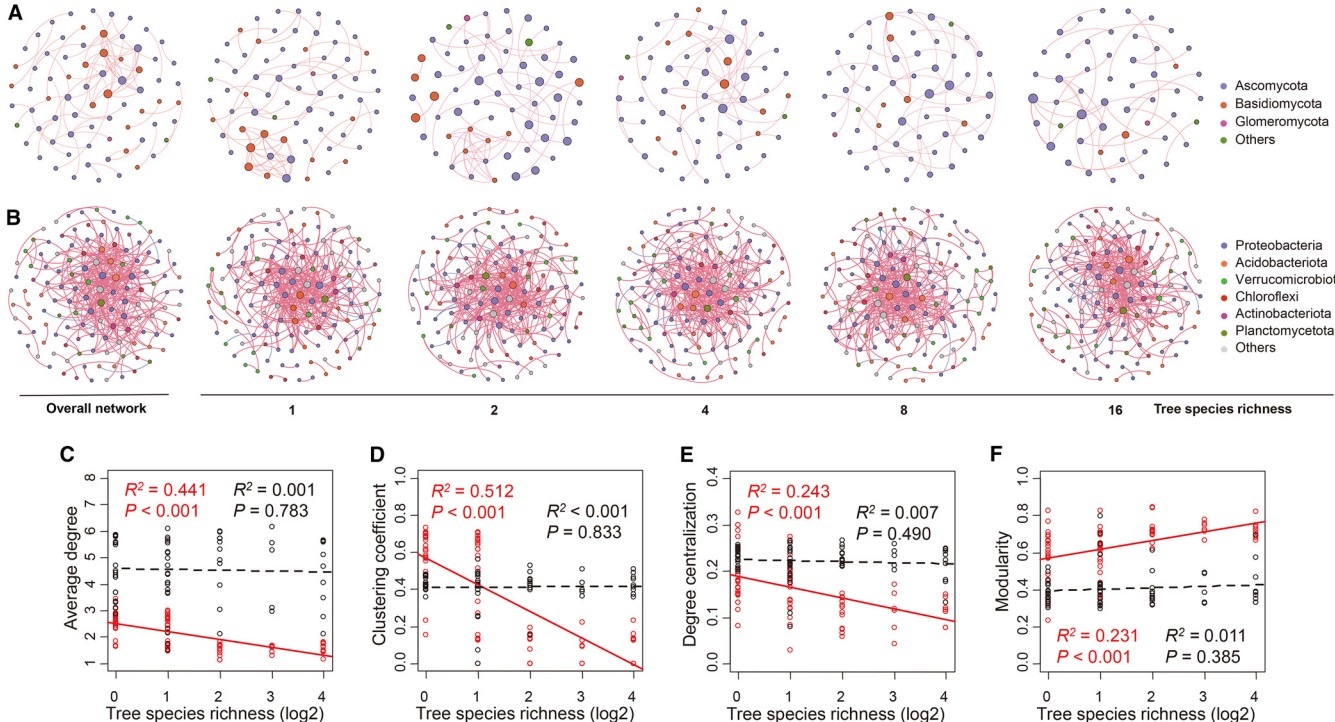

**FIG 6** Architecture and features of the fungal and bacterial co-occurrence networks in different tree species richness classes. (A) The overall fungal co-occurrence network and changes in fungal co-occurrence networks along the tree species richness gradient. (B) The overall bacterial co-occurrence network and changes in bacterial co-occurrence networks along the tree species richness gradient. Positive and negative correlations are indicated by red and blue lines, respectively. The size of each node is proportional to the number of connections (that is, degree). The fungal phyla represent < 1% of the total reads of fungi and fungi not identified to phylum level were all assigned to "Others". The bacterial phyla represent < 5% of the total reads of bacteria and bacteria not identified to phylum level were all assigned to "Others". (C to F) Correlations between tree species richness and topological features of fungal and bacterial co-occurrence networks. Red, fungi. Black, bacteria.

between these topological features and tree species richness were observed in the bacterial co-occurrence network (Fig. 6C to F). These results suggest that plants play more important roles in shaping the co-occurrence network structure of fungi than of bacteria.

## DISCUSSION

The first hypothesis is partially supported by our findings that plants had a stronger effect on the fungal than bacterial community compositions and had no significant effect on fungal and bacterial diversities. Similarly, some previous studies found that plant diversity and/or community composition significantly influenced fungal but not bacterial community compositions in grassland (11, 20–23), oak savannas (25), temperate forest (26, 27), subtropical forest (29), and tropical forest (16, 30) ecosystems. The different effects of plants on fungal and bacterial community compositions may be because fungi are linked more tightly to plants than are bacteria because some fungi can form biotrophic interactions with trees and take the form of root symbionts, endophytes, and pathogens (30–32), but bacteria tend to have a less direct connection to tree roots (62). Furthermore, fungi are more directly dependent on plant-derived resources, such as plant litter and root exudates, than bacteria because fungal decomposers can decompose recalcitrant organic materials (e.g., lignin and cellulose) from the plants (33, 34). In contrast, bacteria mainly utilize the products (e.g., water-soluble sugars and phenolic compounds) released during this process (35, 36), although some bacteria, such as *Nocardia*, *Rhodococcus,* and *Streptomyces viridosporus*, were able to degrade lignin (37–39). Another possibility is that the fungal community responds more quickly to changes in the plant community than the bacterial community (52). In addition, we also found that fungal and bacterial community compositions were

affected by soil variables (pH, carbon: nitrogen ratio [C: N] or phosphorus), and spatial distance, as reported in some previous studies (7–10).

Furthermore, the no significant effect of plants on bacterial diversity in this study is consistent with some previous studies (11–13, 15, 16). However, although we found that plants did not affect soil fungal diversity, a positive correlation between plant species diversity and soil fungal diversity was found in natural forests with a stand age of over 50 years (27, 30). This difference may be because our biodiversity experiment (8 years) is established shortly, or the changes in fungal diversity are obscured by soil legacy effects of the former vegetation (63). In addition, we noticed that the microbial community was analyzed from bulk soil, while we collected soil samples near the trunk of trees. Furthermore, rarefaction curves of the observed OTU richness showed that our results were obtained based on the data analysis of part microbial OTUs existed in soil. Therefore, to better make clear the plant diversity effect on soil microbes, rhizosphere soil which directly interacts with plants, and more samples with deeper sequencing should be used in a future study.

In agreement with our second hypothesis, we found that plants influenced fungal rather than bacterial co-occurrence network structures. Plants may regulate the interactions within the fungal community but not that of the bacterial community by determining the quantity and quality of plant litter and root exudates and by modifying microhabitats (1, 64) because plants have a stronger relationship with fungi than bacteria, as mentioned above. In addition, we also found that the fungal co-occurrence network structure was affected by soil C:N, whereas the bacterial co-occurrence network structure was influenced by soil pH and spatial distance. Similar results were found in previous studies (46, 49). The effect of soil C:N suggests that a shift in soil nutrient status could directly modulate the interactions of the fungal community. The soil pH may affect the interactions of bacteria because pH is very important in determining the variation in bacterial community structure (65, 66) through changes in soil nutrient solubility (67). The effect of spatial distance may be due to the dominance of dispersal limitation in bacterial community assembly (7, 10).

Furthermore, in the fungal rather than bacterial co-occurrence networks, the average degree, degree centralization, and clustering coefficient significantly decreased, but the modularity significantly increased with increasing tree species richness in this study. Our results indicated that interaction intensities between fungal species but not between bacterial species decreased with increasing tree species richness because previous studies suggested that average degree, clustering coefficient, and degree centralization could reflect interaction intensity between species in the co-occurrence network (48, 50, 68). In addition, some studies have interpreted modules as niches (69, 70). The increased tree species richness may lead to there being more niches and stronger niche differentiation, possibly resulting in weaker interspecific interactions in the fungal community but not in the bacterial community because our results revealed that the habitat niche breadth of fungi was always lower than bacteria and decreased with increasing tree species richness (Fig. S8). Furthermore, the higher modularity values may be linked to higher resource availability and habitat complexity for fungi when there is high compared to low tree species richness (64, 71) because modularity was proposed to reflect habitat heterogeneity and divergent selection regimes (72). We speculated the modular organization might enhance the whole network stability, especially by buffering cascades of extinction (70–72). Therefore, plant diversity loss may affect the stability of the fungal co-occurrence network, but not that of the bacterial co-occurrence network.

This study revealed the differences in community assembly and potential drivers of the co-occurrence network structure of soil fungi and bacteria in the subtropical tree diversity experiment. Plant species diversity and composition had significant effects on the community of fungi, but not that of bacteria. In addition to soil, the fungal co-occurrence network was influenced by plants, but the bacterial network was affected by spatial distance. Furthermore, the changes in topological features of the fungal co-occurrence network, but not those of the bacterial co-occurrence network with

increasing tree species richness suggest that the stability of the fungal co-occurrence network rather than that of bacterial co-occurrence network is easily disturbed by plant diversity loss. The study highlights that the community assembly and potential drivers of the co-occurrence network structure of soil fungi and bacteria differ and that plants play more important roles in shaping community assembly and interactions of fungi than of bacteria in our subtropical tree diversity experiment.

## MATERIALS AND METHODS

**Study site and sampling.** The study was conducted at site A in the Biodiversity-Ecosystem Functioning Experiment China (BEF-China), Xingangshan, Jiangxi Province, southeast China (29.08°–29.11°N, 117.90°–117.93°E; 105 to 275 m above sea level). The site is characterized by a subtropical climate, with an annual mean temperature of 16.7°C and annual mean precipitation of 1821 mm (54). After clear-cutting *Pinus massoniana* and *Cunninghamia lanceolata*, a pool of 24 woody plant species native to the regional broadleaved forest was used, in 2009, to create a plantation with different tree species richness classes (54). Briefly, the site covers a total of 271 plots with monocultures and mixtures of 2, 4, 8, 16, and 24 tree species and different shrub addition treatments. Each plot is 25.8 m × 25.8 m in size (Chinese area unit of 1 mu) and planted with 400 seedling individuals (between 1 and 2 years) arranged in a rectangular 20 × 20 grid with 1.29 m spacing between rows and columns.

During September 2017, we selected plots, including 24 tree species at site A of the BEF-China with monocultures and mixtures of 2, 4, 8, and 16 tree species (24-species mixtures were excluded because there were only two replicates). We selected 16 trees (covering all tree species) evenly distributed in each plot and collected one soil core (3.5 cm in diameter, 20 cm in depth) about 0.5 m apart from the trunk of each tree individual after removing plant litter. A total of 16 soil cores were collected and mixed to create one composite sample in each plot. In total, 71 soil samples were collected from 71 plots, including 22 plots with monocultures (two plots were excluded due to high tree mortality), 22 plots with 2-species mixtures (two plots were excluded due to high tree mortality), 12 plots with 4-species mixtures, six plots with 8-species mixtures and nine plots with 16-species mixtures (Table S9). The soil samples were immediately passed through a 2 mm sieve to remove roots and debris and transported to the laboratory in an icebox. To avoid soil contamination between samples, the sieve was disinfected using 75% ethanol after the sieving process for each soil sample. Subsamples were stored at −80°C for DNA extraction, and the remaining subsamples were air-dried for soil physicochemical property analysis.

**Soil parameters and tree volumes.** Soil pH was determined using dried soil mixed with 1 M KCl at a 1:2.5 ratio (wt/vol) using a FiveEasy pH meter (Mettler Toledo, Zurich, Switzerland). Total C and nitrogen (N) were measured by direct combustion using a Vario EL III C/N Element Analyzer (Elementar Analysensysteme GmbH, Germany). Total phosphorus (P) was measured by an inductively coupled plasma spectrometer (iCAP 6300, Thermo Fisher Scientific, Wilmington, USA) after digestion by boiling 0.2 g soil in a solution (5:3) of $HNO_3$ and $HClO_4$ for 75 min (73). Stand-level tree volume (as plant productivity) was determined by tree basal diameter and height in allometric equations for the 16 central trees in each plot, which was corrected by conversion factors determined as total tree volume divided by cylindrical volume (55). Information about soil parameters and tree volumes at different levels of tree species richness is given in Table S10.

**Molecular analysis.** DNA was extracted from 0.25 g of each frozen soil sample using a PowerSoil DNA isolation kit (MoBio Laboratories, Inc. USA). DNA quality and quantity of each sample were measured with a NanoDrop ND-1000 Spectrophotometer (Thermo Scientific, Wilmington, USA). The DNA concentration ranged from 50 to 128 ng/$\mu$L among 71 soil samples. The fungal ITS2 region of the rRNA genes was amplified using primers 5.8SFun (forward) (5′-AACTTTYRRCAAYGGATCWCT-3′) and ITS4Fun (reverse) (5′-AGCCTCCGCTTATTGATATGCTTAART-3′) (74) linked with 12 base barcodes for sample distinction. The PCR mixture (25 $\mu$L) contained 2.5 $\mu$L 10 × buffer, 25 mM $MgSO_4$, 2 mM each dNTP, 10 $\mu$M each primer, 0.5 U KOD-plus-Neo polymerase (Toyobo, Tokyo, Japan), and 10 ng DNA template. Amplifications of ITS2 were performed with an initial denaturation at 94°C for 5 min, followed by 35 cycles of 94°C for 1 min, 56°C for 50 s and 68°C for 1 min, and a final extension at 68°C for 10 min. For bacteria, the V4 hypervariable region of 16S rRNA genes was amplified using primers 515F (5′-GTGCCAGCMGCCGCGGTAA-3′) and 806R (5′-GGACTACVSGGGTATCTAAT-3′) (75) equipped with 12 base barcodes for sample distinction. The 25 $\mu$L reaction solution consisted of 2.5 $\mu$L 10 × buffer ($Mg^{2+}$ plus), 2.5 mM each dNTP, 10 $\mu$M each primer, 1 U Taq DNA polymerase (TaKaRa, Kyoto, Japan), and 10 ng of template DNA. Thermal cycling conditions were as follows: 94°C for 5 min, 35 cycles of 95°C for 50 s, 56°C for 50 s and 72°C for 1 min, followed by 72°C for 10 min. Three replicate PCR products of each sample were pooled and purified using an E.Z.N.A Gel Extraction kit (Omega Bio-Tek, GA, USA) according to the manufacturer's instructions. Sterile deionized distilled water served as negative controls in all steps of the PCR procedure to test for the presence of contamination in reagents. No bands were observed in any of the negative controls. The purified PCR products were pooled with an equal molar amount (100 ng) from each sample and adjusted to 10 ng/$\mu$L. A sequencing library was constructed by adding an Illumina sequencing adaptor (5′-GATCGGAAGAGCACACGTCTGAACTCCAGTCACATCACGATCTCGTATGCCGTCTTCTGCTTG-3′) to the PCR products using an Illumina TruSeq DNA PCR-Free Library Preparation kit (Illumina, CA, USA) according to the manufacturer's instructions. The library sequencing was performed on the Illumina MiSeq PE300 platform running 2 × 300 base pairs (bp) at the Environmental Genome Platform of Chengdu Institute of Biology, Chinese Academy of Sciences, China.

**Bioinformatics analysis.** Clean ITS2 (fungi) and 16S (bacteria) sequences were generated from raw sequences after quality control using Quantitative Insights into Microbial Ecology 2 (QIIME 2) (76). Primer and barcode sequences were excluded using q2-cutadapt (77). The fungal ITS2 region was extracted using the q2-ITSxpress (78). Denoising, removal of chimeras, and dereplication were performed with the DADA2 (79) pipeline implemented in QIIME2. The denoised ITS2 and 16S amplicon sequence variants (ASVs) were clustered into different operational taxonomic units (OTUs) at a threshold of 97% sequence similarity using the vsearch cluster-features-de-novo (80) in QIIME2.

A representative sequence (the most abundant) of each ITS2 and 16S OTU was selected for searching against the entries in the unified system for the DNA based fungal species linked to the classification (UNITE) database (version 04.02.2020) (81) for fungi and against the SILVA database (release 138.1) (82) for bacteria, via the SINTAX algorithm (83) in VSEARCH version 2.18.0 with a confidence cutoff (*P*) value of 0.65. We then excluded the 16S OTUs classified as Archaea from all the samples (2.7% of the total 16S sequences) in further analysis. To eliminate the effect of heterogeneous sequence numbers among the samples on the fungal and bacterial communities identified, the number of sequences per sample was rarefied to the smallest sequence size for fungi and bacteria, respectively, using the sub.sample command in MOTHUR version 1.33.3 (84).

**Statistical analysis.** All the statistical analyses were conducted in R version 3.5.1 (85). The rarefaction curves for the observed OTUs of fungi and bacteria among tree species richness levels were calculated using the specaccum function in the vegan package (86). Linear models were implemented to reveal the responses of fungal and bacterial diversities to plant and abiotic variables using the lm function in the stats package (85).

The distance matrices of the communities of fungi and bacteria were constructed by calculating dissimilarities with Bray-Curtis (87) and Aitchison (88) methods, respectively. Subsequently, nonmetric multidimensional scaling (NMDS) was carried out to visualize the community dissimilarities of fungi and bacteria at different tree species richness levels using the metaMDS function in the vegan package. Permutational multivariate analysis of variance (PerMANOVA) was used to explore the relative importance of tree species richness on the fungal and bacterial community compositions using the adonis command in the vegan package. Furthermore, pairwise PerMANOVA with a false discovery rate correction of *P* values was performed to compare the differences in fungal and bacterial community compositions among different tree species richness using the pairwise.adonis function in the pairwise Adonis package (89). Variation partitioning analysis was undertaken to evaluate the relative importance of plants (richness, volume, and community), soil, and space on the community composition of fungi and bacteria. The spatial principal coordinate of neighbor matrices (PCNM) vectors with positive eigenvalues were obtained via transformation of geographic distance (latitude and longitude) between any plots using the pcnm command in the vegan package. Tree community eigenvectors were derived from the Bray-Curtis matrix based on principal coordinate analysis (PCoA) using the cmdscale command in the vegan package. Significant variables (tree species richness, tree volume, soil properties, and PCoA and PCNM vectors) were forward selected using the forward.sel command in the adespatial package (90). The variations in fungal and bacterial communities were then partitioned according to the selected significant variables using the varpart function in the vegan package. In addition, distance-based redundancy analysis (db-RDA) was conducted to assess the impacts of plants (richness, volume, and PCoA vectors), soil, and space (PCNM vectors) on the community composition of fungi and bacteria in the vegan package. To avoid the effects of collinearity, hierarchical partitioning analysis was used to acquire the independent explanation of each predictor in the rdacca.hp package (91).

Levins' niche breadth (B) index (92, 93) for fungi and bacteria in different tree species richness was calculated using the niche.width function in the spaa package (94) according to the formula:

$$B_j = {}^1\!\big/ {\textstyle\sum_{i=1}^{N} P_{ij}^2}$$

where $B_j$ represents the habitat niche breadth of OTU *j* in each fungal or bacterial community in different tree species richness; *N* is the total number of OTUs in each fungal or bacterial community in different tree species richness; $P_{ij}$ is the proportion of OTU *j* in the community *i*. A high *B*-value for a given OTU indicates its wide habitat niche breadth. The community-level *B*-value (*Bcom*) was calculated as the average of *B*-values from all the OTUs occurring in each fungal or bacterial community in different tree species richness.

Network analysis was applied to explore co-occurrence patterns of fungal and bacterial communities. The overall co-occurrence networks of fungi and bacteria were constructed using the spiec.easi command in the SpiecEasi package (95). The OTUs with relative abundance greater than 0.01% were retained (48). In the analysis, the sample (row) × OTU (column) data matrix (with cell entries indicating the OTU sequences in samples) was used to make network analysis. Data were preprocessed and centered log-ratio (CLR) transformed to ensure compositional robustness and networks were created with the Meinshausen and Buhlmann (MB) network selection method (96) in the SpiecEasi, with a nlambda penalty value of 20 (97). The nodes in this network represented OTUs and the links that connected these nodes represented correlations between OTUs. The topological features average degree, clustering coefficient, degree centralization, and modularity were used in this study. Average degree is a quantification feature indicating the number of direct co-occurrence interactions (68). The clustering coefficient is a measure of the likelihood that the adjacent species of a specific species are connected (48). Degree centralization describes a particular pattern of interaction in which it is close to 1 for a network with a star topology and, in contrast, close to 0 for a network where each species exhibits the same links (68).

Modularity is a measure of the extent to which the network is structured as modules (68). These four topological features were calculated using the igraph package (98). Meanwhile, 1,000 Erdös-Réyni random networks of an equal size were constructed to compare with the topology of the empirical networks (99). Networks were visualized using the interactive platform Gephi (100). Significant PCNM vectors, plant and soil variables were forward-selected for subsequent statistical analyses using the forward.sel command in the adespatial package. After the forward selection procedures, the variations in topological features of fungal and bacterial networks were partitioned using the varpart function in the vegan package.

To explore the relationship between tree species richness and the topological features of the fungal and bacterial networks, we reconstructed subnetworks of fungi and bacteria in different tree species richness classes from the overall networks with the subgraph function in the igraph package. Subnetworks for each soil sample, maintaining OTUs associated with specific samples and all links between them in the overall networks, were generated using the subgraph function in the igraph package, and four topological features for each soil sample were calculated with the igraph package.

**Data availability.** The raw data have been submitted to the Genome Sequence Archive (GSA) in National Genomics Data Center, China National Center for Bioinformation/Beijing Institute of Genomics, Chinese Academy of Sciences under accession number CRA006693.

## SUPPLEMENTAL MATERIAL

Supplemental material is available online only.
**SUPPLEMENTAL FILE 1**, PDF file, 1.1 MB.
**SUPPLEMENTAL FILE 2**, XLSX file, 1 MB.
**SUPPLEMENTAL FILE 3**, XLSX file, 0.8 MB.
**SUPPLEMENTAL FILE 4**, XLSX file, 0.02 MB.

## ACKNOWLEDGMENTS

We thank the staff of the Biodiversity-Ecosystem Functioning Experiment China (BEF-China) in Xingangshan, Jiangxi Province for their assistance in soil sample collection. We are grateful to Helge Bruelheide and Ye Deng for their valuable suggestions on the data analyses and Cheng Gao for helpful comments on the paper.

L.D.G. conceived the idea for the study. L.D.G. and X.C.L. designed the research. Y.L.W., X.C.L., and H.Y.G. collected the samples. H.Y.G. performed the laboratory work. N.N.J., X.C.L., P.P.L., Y.L.W., H.Y., S.L., and H.Y.G. analyzed the data and/or contributed data and advice. H.Y.G. wrote the manuscript. All authors read and approved the final manuscript.

This work was supported financially by the Strategic Priority Research Program of the Chinese Academy of Sciences (grant number XDB31030000) and the National Natural Science Foundation of China (grant number 31700543). We gratefully acknowledge the International Research Training Group TreeDì jointly funded by the Deutsche Forschungsgemeinschaft (DFG, German Research Foundation) grant number 319936945/GRK2324 and the University of Chinese Academy of Science (UCAS).

We declare no conflict of interest.

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
