## [Reviewer comments · Microbiology Spectrum]

Microbiology Spectrum

Plants Play Stronger Effects on Soil Fungal Than Bacterial Communities and Co-occurrence Network Structures in a Subtropical Tree Diversity Experiment

Huiyun Gan, Xingchun Li, Yonglong Wang, Pengpeng Lü, Niuniu Ji, Hui Yao, Shan Li, and Liangdong Guo

Corresponding Author(s): Liangdong Guo, Institute of Microbiology, Chinese Academy of Sciences; College of Life Sciences, University of Chinese Academy of Sciences

Review Timeline:

Submission Date:	January 17, 2022
Editorial Decision:	February 21, 2022
Revision Received:	March 2, 2022
Accepted:	March 11, 2022

Editor: Cheng Gao

Reviewer(s): The reviewers have opted to remain anonymous.

Transaction Report:

DOI: <https://doi.org/10.1128/spectrum.00134-22>

February 21, 2022

Prof. Liangdong Guo
Institute of Microbiology, Chinese Academy of Sciences; College of Life Sciences, University of Chinese Academy of Sciences
No. 3 1st Beichen West Rd.
Beijing, Beijing China
China

Re: Spectrum00134-22 (Plants Play Stronger Effects on Soil Fungal Than Bacterial Communities and Co-occurrence Network Structures in a Subtropical Tree Diversity Experiment)

Dear Prof. Liangdong Guo:

Thank you for submitting your manuscript to Microbiology Spectrum. Your manuscript has been reviewed by two reviewers. After careful consideration, we invite a resubmission. When submitting the revised version of your paper, please provide (1) point-by-point responses to the issues raised by the reviewers as file type "Response to Reviewers," not in your cover letter, and (2) a PDF file that indicates the changes from the original submission (by highlighting or underlining the changes) as file type "Marked Up Manuscript - For Review Only". Please use this link to submit your revised manuscript - we strongly recommend that you submit your paper within the next 60 days or reach out to me. Detailed instructions on submitting your revised paper are below.

Link Not Available

Sincerely,

Cheng Gao

Journals Department
Reviewer comments:

Reviewer #1 (Comments for the Author):

Gan et al. investigated the soil fungal and bacterial diversity, community composition and co-occurrence network structures in a plant diversity gradient (1-16 species) experimental in a subtropical forest ecosystem, using amplicon sequencing targeting the ITS2 (fungi) and 16S rRNA gene V4 hypervariable (bacteria) regions, respectively. The results showed that plant (tree) species richness had little effects on fungal and bacterial diversities, but both plant diversity and community composition exerted significant impacts on only fungal community composition. Moreover, fungal rather than bacterial co-occurrence network properties were significantly influenced by tree species richness. Different environmental factors were identified to affect fungal

(tree community and soil C/N) and bacterial (soil pH and spatial distance) co-occurrence network structure. Overall, I feel this work focused on an important topic of microbial community ecology, that is, how soil microbial diversity and community would shift under different plant diversity levels, which would occur under global environment changes scenarios in future. The experiment was well designed, and the methods adopted and data analyses were solid. I also think the manuscript was well written and was easy to follow. However, I still have some comments or unclear things which would be considered by the authors, and I hope these suggestions may good for improving the manuscript.

1. The Bray-Curtis distance is commonly used, although it does not consider the compositional nature of sequencing data. The Aitchison distance is a viable compositional alternative. How did the authors handle the compositional nature of the data (Alteio et al., 2021)? [Alteio et al., 2021 A critical perspective on interpreting amplicon sequencing data in soil ecological research. SBB, <https://doi.org/10.1016/j.soilbio.2021.108357>]

2. A query is 'did the tree volume exert any effect on fungal and bacterial communities?' Maybe yes?

3. For the introduction section, there are some latest references which can be included for introducing the plant diversity effect on soil microbes. For example, Schmid et al. 2021, J Ecol. and Prada-Salcedo et al. 2021, Mol Ecol.

Schmid, M. W., van Moorsel, S. J., Hahl, et al. (2021). Effects of plant community history, soil legacy and plant diversity on soil microbial communities. *Journal of Ecology*.

Prada-Salcedo, L. D., Goldmann, K., Heintz-Buschart, A., et al. (2021). Fungal guilds and soil functionality respond to tree community traits rather than to tree diversity in European forests. *Molecular Ecology*, 30(2), 572-591.

4. Some minor comments listed below,

1) Lines 47, 125, "that" change into "those".

2) L49-50, I do not think the presentation of "the subtropical plant diversity forest" is proper. Please rephrase this sentence.

3) Lines 66-67, soil fungal and bacterial community diversities and compositions could be affected by abiotic factors.

4) For fig. 1, I am confused that does this figure show the results of correlation analyses between two parameters?

5) Fig. 4, I suggest that you should state that plant factor is an integration of tree species richness and tree community composition.

Reviewer #2 (Comments for the Author):

The authors of "Plants Play Stronger Effects on Soil Fungal Than Bacterial Communities and Co-occurrence Network Structures in a Subtropical Tree Diversity Experiment" used high-throughput sequencing technology to study the community composition differences and potential driving factors of soil fungi and bacterial symbiotic network structure in a subtropical plant diversity forest. Through the co-occurrence network analysis, it was concluded that the influence of plants on the co-occurrence network structure of soil fungi was greater than that of bacteria. The work is very interesting, but some questions and suggestions for improvement are put forward.

Here my point to point comments :

1. Lines 148-150 of your Manuscript Draft: With the increase of the number of samples, the fungal species rarefaction curves of the microbial community did not reach the saturation stage, which indicated that the microorganisms in the community did not capture most of the fungal members. Will the resulting analysis lose some authenticity ? An explanation is suggested.

2. Lines 173-175 of your Manuscript Draft: Here the forward regression method is used to select important variables, why not consider all-subsets regression? In addition, variable selection may appear to choose a model that is mathematically significant but biologically meaningless. For example, here the bacteria only chooses Space and soil factors, but it does not mean that the tree community has no effect on bacterial composition. How to avoid this situation?

3. Lines 227-229 of your Manuscript Draft: This inference is interesting, but other references are needed to show this relationship.

4. Lines 231-233 of your Manuscript Draft: The above analysis results do not strongly support the conclusion, and it is suggested to revise it to a more appropriate summary.

5. Lines 267-269 of your Manuscript Draft: Why not consider further analysis of bacterial/fungal niche breadth under different tree species richness to increase the power of this inference?

6. Lines 273-276 of your Manuscript Draft: Why not consider further analysis of bacterial/fungal network stability under different tree species richness to increase the convincing power of this inference?

7. Lines 422-423 of your Manuscript Draft: It is suggested to supplement the method of network construction and the threshold of screening.

Staff Comments:

Preparing Revision Guidelines

Please return the manuscript within 60 days; if you cannot complete the modification within this time period, please contact me. If you do not wish to modify the manuscript and prefer to submit it to another journal, please notify me of your decision immediately so that the manuscript may be formally withdrawn from consideration by Microbiology Spectrum.

Reviewer #1 (Comments for the Author):

■ Gan et al. investigated the soil fungal and bacterial diversity, community composition and co-occurrence network structures in a plant diversity gradient (1-16 species) experimental in a subtropical forest ecosystem, using amplicon sequencing targeting the ITS2 (fungi) and 16S rRNA gene V4 hypervariable (bacteria) regions, respectively. The results showed that plant (tree) species richness had little effects on fungal and bacterial diversities, but both plant diversity and community composition exerted significant impacts on only fungal community composition. Moreover, fungal rather than bacterial co-occurrence network properties were significantly influenced by tree species richness. Different environmental factors were identified to affect fungal (tree community and soil C/N) and bacterial (soil pH and spatial distance) co-occurrence network structure. Overall, I feel this work focused on an important topic of microbial community ecology, that is, how soil microbial diversity and community would shift under different plant diversity levels, which would occur under global environment changes scenarios in future. The experiment was well designed, and the methods adopted and data analyses were solid. I also think the manuscript was well written and was easy to follow. However, I still have some comments or unclear things which would be considered by the authors, and I hope these suggestions may good for improving the manuscript.

Response: Thank you very much for your positive comments. We have made relevant revisions (marked in red) in the revised manuscript.

■ The Bray-Curtis distance is commonly used, although it does not consider the compositional nature of sequencing data. The Aitchison distance is a viable compositional alternative. How did the authors handle the compositional nature of the data (Alteio et al., 2021)? [Alteio et al., 2021 A critical perspective on interpreting amplicon sequencing data in soil ecological research. SBB, <https://doi.org/10.1016/j.soilbio.2021.108357>]

Response: Thank you for your valuable suggestions. We constructed the distance matrices of the communities of fungi and bacteria by calculating dissimilarities with Aitchison method (line 414). Similar results were found in the Permutational multivariate analysis of variance (PerMANOVA), non-metric multidimensional scaling (NMDS), variation partitioning and hierarchical partitioning analyses, using Aitchison matrices of bacterial and fungal communities (lines 183-185; Fig. S3 to S5, Table S6 and S7).

■ A query is 'did the tree volume exert any effect on fungal and bacterial communities?' Maybe yes?

Response: Thanks for your valuable suggestion. We used linear models and variation partitioning analysis to check the responses of fungal and bacterial communities to the tree volume. The results indicated that the tree volume had no significant effects on fungal and bacterial richness (Table S3) and community composition (fungi: $F=1.3808$, $P=0.061$; bacteria: $F=0.8635$, $P=0.672$).

■ For the introduction section, there are some latest references which can be included for introducing the plant diversity effect on soil microbes. For example, Schmid et al. 2021, J Ecol. and Prada - Salcedo et al. 2021, Mol Ecol.

Schmid, M. W., van Moorsel, S. J., Hahl, et al. (2021). Effects of plant community history, soil legacy and plant diversity on soil microbial communities. Journal of Ecology.

Prada - Salcedo, L. D., Goldmann, K., Heintz - Buschart, A., et al. (2021). Fungal guilds and soil functionality respond to tree community traits rather than to tree diversity in European forests. Molecular Ecology, 30(2), 572-591.

Response: Thank you for your recommendation. We have read two papers and added them to the introduction in lines 74-75.

Some minor comments listed below,

- Lines 47, 125, "that" change into "those".

Response: Corrected (lines 47 and 125).

- L49-50, I do not think the presentation of "the subtropical plant diversity forest" is proper. Please rephrase this sentence.

Response: Thanks for your comments. We have changed “the subtropical plant diversity forest” into “the subtropical tree diversity experiment” and revised all similar sentences throughout based on your comment.

- Lines 66-67, soil fungal and bacterial community diversities and compositions could be affected by abiotic factors.

Response: Corrected (lines 66-67).

- For fig. 1, I am confused that does this figure show the results of correlation analyses between two parameters?

Response: Yes. We showed the results of correlation analyses between fungal and bacterial richness, Shannon diversity index and Simpson diversity index and tree species richness in Fig. 1. So, we changed the title of figure 1 (line 924).

- Fig. 4, I suggest that you should state that plant factor is an integration of tree species richness and tree community composition.

Response: Thanks for your valuable suggestion. We have stated it in figure legend (line 956).

Reviewer #2 (Comments for the Author):

- The authors of "Plants Play Stronger Effects on Soil Fungal Than Bacterial Communities and Co-occurrence Network Structures in a Subtropical Tree Diversity Experiment" used high-throughput sequencing technology to study the community

composition differences and potential driving factors of soil fungi and bacterial symbiotic network structure in a subtropical plant diversity forest. Through the co-occurrence network analysis, it was concluded that the influence of plants on the co-occurrence network structure of soil fungi was greater than that of bacteria. The work is very interesting, but some questions and suggestions for improvement are put forward.

Response: Thank you very much for your positive comments. We have made revisions (marked in red) according to your useful comments and suggestions.

Here my point to point comments:

■ Lines 148-150 of your Manuscript Draft: With the increase of the number of samples, the fungal species rarefaction curves of the microbial community did not reach the saturation stage, which indicated that the microorganisms in the community did not capture most of the fungal members. Will the resulting analysis lose some authenticity? An explanation is suggested.

Response: Thank you for your comments. We have explained it in lines 252-256.

■ Lines 173-175 of your Manuscript Draft: Here the forward regression method is used to select important variables, why not consider all-subsets regression? In addition, variable selection may appear to choose a model that is mathematically significant but biologically meaningless. For example, here the bacteria only chooses Space and soil factors, but it does not mean that the tree community has no effect on bacterial composition. How to avoid this situation?

Response: Thank you for your suggestions. Distance based redundancy analysis (db-RDA) was conducted to assess the impacts of plants (richness, volume, and PCoA vectors), soil and space (PCNM vectors) on the community composition of fungi and bacteria in the vegan package (lines 436-438). To avoid the effects of collinearity, hierarchical partitioning analysis was used to acquire the independent explanation of each predictor in the rdacca.hp package (lines 438-440). The rdacca.hp package allow us to put all the predictors during analysis without screening process. We got similar

results compared with the results conducted by variation partitioning (lines 176-183 and 942-950; Fig. 4 and Table S5).

■ Lines 227-229 of your Manuscript Draft: This inference is interesting, but other references are needed to show this relationship.

Response: Thank you for your comments. We have included one reference to show this relationship (line 240).

■ Lines 231-233 of your Manuscript Draft: The above analysis results do not strongly support the conclusion, and it is suggested to revise it to a more appropriate summary.

Response: Thanks for your valuable suggestion. We agree with your idea that it could be not strongly support the conclusion. So, we deleted it.

■ Lines 267-269 of your Manuscript Draft: Why not consider further analysis of bacterial/fungal niche breadth under different tree species richness to increase the power of this inference?

Response: Thanks for your suggestions. We analyzed the bacterial and fungal niche breadth under different tree species richness, added the method of calculating the Levins' niche breadth index (lines 441-450) and put the results in the supplemental file 1 (FIG S8). We found that the habitat niche breadth of fungi was always lower than bacteria and decreased with increasing tree species richness. It showed that stronger niche differentiation in the fungal community than in the bacterial community with increasing tree species richness, possibly resulting in weaker interspecific interactions in the fungal community but not in the bacterial community (lines 282-283).

■ Lines 273-276 of your Manuscript Draft: Why not consider further analysis of bacterial/fungal network stability under different tree species richness to increase the convincing power of this inference?

Response: We analyzed the robustness of fungal and bacterial networks in different tree species richness and found the robustness of fungal but not bacterial networks increased with increasing tree species richness, though the effect was not significant (see Fig. 1 below). So, we revised that inference (lines 287-288).

Fig. 1 The fungal (red) and bacterial (black) co-occurrence network robustness in different tree species richness.

■ Lines 422-423 of your Manuscript Draft: It is suggested to supplement the method of network construction and the threshold of screening.

Response: Thanks for your comments. We have supplemented the method of network construction and the threshold of screening in lines 454-459.

March 11, 2022

Prof. Liangdong Guo
Institute of Microbiology, Chinese Academy of Sciences; College of Life Sciences, University of Chinese Academy of Sciences
No. 3 1st Beichen West Rd.
Beijing, Beijing China
China

Re: Spectrum00134-22R1 (Plants Play Stronger Effects on Soil Fungal Than Bacterial Communities and Co-occurrence Network Structures in a Subtropical Tree Diversity Experiment)

Dear Prof. Liangdong Guo:

Your manuscript has been accepted, and I am forwarding it to the ASM Journals Department for publication. You will be notified when your proofs are ready to be viewed.

Sincerely,

Cheng Gao
Editor, Microbiology Spectrum

Journals Department
SUPPLEMENTAL FILE 2: Accept
SUPPLEMENTAL FILE 3: Accept
SUPPLEMENTAL FILE 1: Accept
SUPPLEMENTAL FILE 4: Accept